# Combinatorial Capacity of modReLU Complex Networks: VC-Dimension Bounds and Lower Limits

**Mehmet Altunören**                                                        *mehmet.altunoren@sabanciuniv.edu*
*Sabancı University*

**Reviewed on OpenReview:** *https://openreview.net/forum?id=jfeJnfST36*

## Abstract

Complex-valued neural networks (CVNNs) are increasingly used in settings where both magnitude and phase of the signal carry information; see, e.g., (12; 13; 15; 17). In particular, deep networks with the modReLU activation (2; 20) have been used extensively in applications such as MRI reconstruction, radar, and complex-valued time-series modeling. While approximation properties of such networks have recently been analyzed in detail (8), their statistical capacity in the sense of VC-dimension has not, to the best of our knowledge, been studied. In this paper we formalize a natural class of fully connected deep complex-valued networks with modReLU activation and real sign output. Via the standard identification $\mathbb{C}^d \cong \mathbb{R}^{2d}$, we view these models as binary classifiers on $\mathbb{R}^{2d}$. Let $W$ denote the total number of real-valued trainable parameters, including the real and imaginary parts of all weights and biases as well as the real modReLU bias parameters. Using tools from real algebraic geometry and a VC-dimension bound for semi-algebraic concept classes due to Goldberg and Jerrum (10), together with quantitative bounds for quantifier elimination (5), we prove that for any architecture with $W$ parameters and depth $L$, the VC-dimension of the corresponding hypothesis class is at most on the order of $W^2 \log W$, with a universal constant independent of the particular architecture. On the other hand, by restricting to real inputs and parameters and exploiting results of Harvey, Liaw, and Mehrabian (11) and Bartlett et al. (4) on deep networks with piecewise-linear activations, we obtain lower bounds of order $WL \log(W/L)$ for suitable depth-$L$ architectures within the modReLU class. Thus the VC-dimension of these networks grows at least linearly in both $W$ and $L$, and at most quadratically in $W$ up to a logarithmic factor. Closing this gap is an interesting open problem.

## 1 Introduction

In many signal-processing problems the natural data space is complex-valued. Examples include MRI, radar, wireless communication channels, and certain Fourier or wavelet representations of time series. For such problems, complex-valued neural networks (CVNNs) have been proposed as a more natural alternative to real networks; see, for instance, (12; 13; 14; 20; 15; 17) and the references therein. In these architectures all weights, biases, and activations are complex, and the network computes a map from $\mathbb{C}^d$ to $\mathbb{C}^k$.

**Other complex-valued activations.** modReLU is only one of several activation functions used in CVNNs. Beyond amplitude–phase constructions such as modReLU (2; 20), common choices include *split* activations that apply a real nonlinearity separately to $\Re(z)$ and $\Im(z)$ (often called *CReLU* / split-ReLU), quadrant-gated variants such as zReLU, and learned or nonparametric complex activations (e.g., kernel-based complex activation mechanisms) (15; 16). These alternatives are widely used in practice and sometimes preferred for optimization or implementation reasons (e.g., avoiding explicit normalization by $|z|$).

Recently, there has been considerable progress on the approximation theory of CVNNs. In particular, Geuchen and Voigtlaender (8) established explicit quantitative error bounds for approximating smooth func-

tions on compact subsets of $\mathbb{C}^d$ by complex-valued networks with the modReLU activation function (20), and showed that the rates are optimal up to logarithmic factors. Follow-up work has treated phenomena such as depth separation and universality for narrow complex networks (9; 14). A recent survey by Bassey, Qian, and Li (15) gives a broad overview of complex activations, Wirtinger calculus, and optimization aspects.

In contrast, the statistical learning theory of CVNNs is still largely undeveloped. For real networks, a rich body of work has characterized the VC-dimension and pseudo-dimension of various architectures since the classical paper of Baum and Haussler (6) and the monograph of Anthony and Bartlett (1). For example, deep ReLU networks with $W$ real parameters and depth $L$ are now known to have VC-dimension on the order of $WL \log W$, and this dependence is essentially tight (11; 4). Related results characterize the pseudo-dimension of real networks in regression settings (18; 22).

A natural question is whether these nearly tight results for real networks can be transferred to complex-valued architectures by using the standard identification $\mathbb{C}^d \cong \mathbb{R}^{2d}$ (sending $x + iy$ to $(x, y)$ coordinate-wise). For networks with activations that remain piecewise-polynomial after this identification, one can indeed hope to import the classical region-counting and sign-pattern arguments; this includes, for instance, split activations such as CReLU, which avoid normalization by $|z|$ and hence typically admit a more direct reduction to the real piecewise-linear setting. For modReLU networks, however, such a transfer is *not* available as a black box. The sharp $WL \log W$ analyses for deep real networks in (11; 4) proceed by exploiting that the network computes a *piecewise-polynomial* (in particular, piecewise-linear) map of the input, with a partition induced by activation gates; this structure enables counting the number of distinct affine regions and hence the number of labelings the network can realize. In contrast, when modReLU is written in real coordinates $z = x + iy$, its active branch contains the normalization factor $z/|z|$ with $|z| = \sqrt{x^2 + y^2}$, i.e., expressions involving division by $\sqrt{x^2 + y^2}$. Consequently, the real-coordinate representation of modReLU is not piecewise-polynomial in $(x, y)$ in the sense required by these arguments, and the existing $\Theta(WL \log W)$ upper-bound proofs for piecewise-linear networks cannot be applied directly to modReLU architectures.

**Why modReLU merits a dedicated analysis.** Among complex-valued activations, modReLU occupies a distinctive position that makes it both practically important and theoretically challenging. On the practical side, modReLU is the standard nonlinearity in several influential CVNN architectures for MRI reconstruction, radar signal processing, and unitary recurrent networks (2; 20; 17), and its phase-preserving property—the activation rescales the magnitude while leaving the phase of its input unchanged—is a key inductive bias in domains where phase information carries physical meaning (e.g., coherent imaging and array signal processing). On the theoretical side, modReLU is the simplest widely-used complex activation whose real-coordinate representation is *genuinely non-piecewise-polynomial*: the normalization $z/|z|$ introduces algebraic (square-root) nonlinearities that prevent a direct reduction to the well-understood piecewise-linear setting. Split activations such as CReLU, by contrast, reduce immediately to real piecewise-linear networks after identifying $\mathbb{C}^d \simeq \mathbb{R}^{2d}$, and thus inherit the classical $\Theta(WL \log W)$ bounds with only routine bookkeeping. Studying modReLU therefore requires genuinely new proof techniques—the semi-algebraic lifting and quantifier-elimination strategy developed here—which are likely to be relevant for other activations with similar algebraic structure (e.g., phase-dependent gating mechanisms or activations involving $|z|$ in a non-polynomial way).

Our contribution is to show that, despite this obstruction, the decision regions of modReLU networks can still be controlled using *semi-algebraic* methods after a suitable lifting. By introducing auxiliary radius variables $r$ satisfying $r^2 = x^2 + y^2$, the modReLU gate and the full forward pass admit an existential description by polynomial equalities and inequalities of bounded degree. Projecting out the auxiliary variables yields a semi-algebraic description of the decision set via Tarski–Seidenberg (quantifier elimination) (19; 5). This reduction allows us to invoke the general VC-dimension bound for semi-algebraic concept classes due to Goldberg and Jerrum (10), together with quantitative bounds for quantifier elimination (5), to obtain parameter-count-dependent VC-dimension upper bounds for modReLU complex networks. The price of using a generic projection/quantifier-elimination step is an additional factor in the resulting description complexity, which is reflected in the $O(W^2 \log W)$ bound proved in this work.

**Implications for other activations.** The same high-level semi-algebraic strategy can be applied to other complex activations whose input–output relation can be expressed (possibly after introducing auxiliary variables) by polynomial equalities and inequalities of bounded degree. In particular, for activations that are piecewise-polynomial after identifying $\mathbb{C}^d \simeq \mathbb{R}^{2d}$ (such as split activations), one expects that sharper $O(WL \log W)$-type upper bounds may be attainable by adapting the classical real-network arguments, without incurring the additional blow-up from generic quantifier elimination. For example, for split activations such as CReLU/split-ReLU, after identifying $\mathbb{C}^d \simeq \mathbb{R}^{2d}$ the network becomes a standard real piecewise-linear network with a constant number of pieces, so the known $\Theta(WL \log W)$ VC-dimension bounds (up to constant-factor changes in parameter counting) apply directly (11; 4). A systematic VC-dimension comparison across the landscape of complex activations (including learned/nonparametric ones (16)) is an interesting direction for future work.

Beyond serving as a combinatorial complexity measure, the VC-dimension controls the growth function (via Sauer–Shelah type bounds) and yields classical sample-complexity and uniform-convergence guarantees for ERM-style learning (21; 1). Thus, bounding the VC-dimension of CVNNs provides a principled way to compare their statistical capacity to real-valued architectures and complements existing norm-based generalization analyses (7).

For complex-valued networks there are, to the best of our knowledge, no analogous capacity results in terms of VC-dimension. The only global complexity bounds we are aware of are recent norm-based generalization bounds that control the generalization gap via a spectral-complexity measure for CVNNs (7), which is a different notion from combinatorial capacity. The main purpose of this work is to begin closing this gap by providing VC-dimension bounds for a widely used complex-valued architecture family.

**Contributions.** At a high level, the contributions of this paper are as follows.

- We formalize a natural class of fully connected deep complex-valued networks with modReLU activation and real sign output. Via the standard identification $\mathbb{C}^d \cong \mathbb{R}^{2d}$, we view these models as binary classifiers on $\mathbb{R}^{2d}$.

- We show that the corresponding decision regions form a semi-algebraic family in $(\theta, x)$ whose description complexity—in terms of the number and degree of defining polynomials—can be bounded in terms of the architecture and the number $W$ of real-valued trainable parameters.

- Using the VC-dimension bound for semi-algebraic concept classes due to Goldberg and Jerrum (10), combined with quantitative bounds on quantifier elimination (5), we derive an upper bound of order $W^2 \log W$ for the VC-dimension of this class.

- By specializing to real inputs and constraining parameters appropriately, we show that our class contains, as a subclass, real-valued networks with a fixed piecewise-linear activation. Results of Harvey, Liaw, and Mehrabian (11) and Bartlett et al. (4) then imply lower bounds of order $WL \log(W/L)$ for suitable architectures, showing that the VC-dimension must grow at least linearly in both $W$ and $L$.

- We discuss how the same approach could be extended to other complex activations and to pseudo-dimension, and outline several open problems, including whether the $W^2 \log W$ upper bound can be sharpened towards $WL \log W$ in the complex-valued setting.

## 2 Preliminaries

We collect notation and background results used in the sequel. Standard references for VC-dimension and statistical learning theory include (21; 1; 22).

**Notation.** We write $\Re(z)$ and $\Im(z)$ for the real and imaginary parts of $z \in \mathbb{C}$. We identify $\mathbb{C}^d$ with $\mathbb{R}^{2d}$ via the bijection $\Psi : \mathbb{R}^{2d} \to \mathbb{C}^d$ defined by

$$\Psi(a_1, b_1, \ldots, a_d, b_d) := (a_1 + ib_1, \ldots, a_d + ib_d).$$

For binary classification we use sign : $\mathbb{R} \to \{+1, -1\}$ with the convention $\text{sign}(t) = +1$ if $t \geq 0$ and $\text{sign}(t) = -1$ if $t < 0$. A network of depth $L$ has $L - 1$ hidden layers and one output layer; we index layers by $\ell = 0, \ldots, L$ with $n_0 = d$ and $n_L = 1$.

**VC-dimension.** Let $\mathcal{H} \subseteq \{+1, -1\}^{\mathcal{X}}$ be a class of binary classifiers on a domain $\mathcal{X}$. A finite set $\{x_1, \ldots, x_m\} \subseteq \mathcal{X}$ is *shattered* by $\mathcal{H}$ if for every labeling $(y_1, \ldots, y_m) \in \{+1, -1\}^m$ there exists $h \in \mathcal{H}$ such that $h(x_i) = y_i$ for all $i$. The *VC-dimension* $\text{VCdim}(\mathcal{H})$ is the largest $m$ such that some set of size $m$ is shattered (or $\infty$ if no such largest $m$ exists) (21; 1).

**Semi-algebraic sets.** A set $S \subset \mathbb{R}^n$ is called *semi-algebraic* if it can be expressed as a finite Boolean combination of polynomial equalities and inequalities in the coordinates $x_1, \ldots, x_n$. A family $\{S_\theta\}_{\theta \in \Theta}$ with $\Theta \subseteq \mathbb{R}^W$ is called semi-algebraic if

$$\{(\theta, x) \in \Theta \times \mathbb{R}^n : x \in S_\theta\}$$

is a semi-algebraic subset of $\mathbb{R}^{W+n}$. A standard closure result is the Tarski–Seidenberg theorem: the image (projection) of a semi-algebraic set under a polynomial map is semi-algebraic (19; 5).

We will use the following general bound on the VC-dimension of semi-algebraic families due to Goldberg and Jerrum (10) (see also (1, Theorem 8.3) and (5, Theorem 7.38)).

**Theorem 1 (Goldberg–Jerrum (10))** *Let $F = \{S_\theta\}_{\theta \in \Theta}$ be a family of subsets of $\mathbb{R}^n$ parameterized by $\Theta \subset \mathbb{R}^W$. Suppose there exist integers $s \geq 1$ and $d \geq 1$ and polynomials $p_1, \ldots, p_s \in \mathbb{R}[U_1, \ldots, U_W, X_1, \ldots, X_n]$ of degree at most $d$ such that for all $\theta \in \Theta$ and $x \in \mathbb{R}^n$,*

$$x \in S_\theta \iff \Phi\big(\text{sign}(p_1(\theta, x)), \ldots, \text{sign}(p_s(\theta, x))\big) = \text{true},$$

*for some fixed Boolean formula $\Phi$. Then there exists a universal constant $C > 0$ such that*

$$\text{VCdim}(F) \leq C W \log(s\,d).$$

**Complex-valued neural networks and modReLU.** Let $d \geq 1$ and $L \geq 1$ be integers. A fully connected feedforward complex-valued network (CVNN) of depth $L$ and input dimension $d$ consists of:

- layer widths $n_0 = d$, $n_1$, $\ldots$, $n_{L-1}$, $n_L = 1$,

- complex weight matrices $W^{(\ell)} \in \mathbb{C}^{n_\ell \times n_{\ell-1}}$ and complex biases $b^{(\ell)} \in \mathbb{C}^{n_\ell}$ for $\ell = 1, \ldots, L$,

- a real modReLU bias parameter $\beta_j^{(\ell)} \in \mathbb{R}$ for each hidden unit $(\ell, j)$ with $\ell = 1, \ldots, L - 1$.

Given an input $z^{(0)} = x \in \mathbb{C}^d$, the network computes recursively for $\ell = 1, \ldots, L - 1$:

$$u^{(\ell)} = W^{(\ell)} z^{(\ell-1)} + b^{(\ell)} \in \mathbb{C}^{n_\ell}, \qquad z_j^{(\ell)} = \sigma_{\text{modR}}\big(u_j^{(\ell)}; \beta_j^{(\ell)}\big), \quad j = 1, \ldots, n_\ell,$$

and the (complex-valued) output is

$$F_\theta(x) = W^{(L)} z^{(L-1)} + b^{(L)} \in \mathbb{C},$$

where $\theta$ denotes the collection of all trainable real parameters (real and imaginary parts of all weights and biases, together with all $\beta_j^{(\ell)}$). The total number of trainable real parameters is therefore

$$W = 2 \sum_{\ell=1}^{L} n_\ell n_{\ell-1} + 2 \sum_{\ell=1}^{L} n_\ell + \sum_{\ell=1}^{L-1} n_\ell.$$

We work with the *modReLU* activation introduced by Arjovsky et al. (2) and later used in deep complex networks by Trabelsi et al. (20) (see also (8)). For $\beta \in \mathbb{R}$ and $z \in \mathbb{C}$, define

$$\sigma_{\text{modR}}(z; \beta) := \begin{cases} 0, & z = 0, \\ 0, & |z| + \beta \leq 0, \\ (|z| + \beta) \dfrac{z}{|z|}, & |z| + \beta > 0, \end{cases} \tag{1}$$

i.e., $\sigma_{\mathrm{modR}}(z; \beta) = \mathrm{ReLU}(|z| + \beta) \cdot \dfrac{z}{|z|}$ with the convention $\dfrac{z}{|z|} = 0$ at $z = 0$ (2; 20). Equivalently, writing $z = re^{i\vartheta}$ with $r = |z|$ and $\vartheta = \arg(z)$ (defined for $r > 0$), one has $\sigma_{\mathrm{modR}}(z; \beta) = 0$ for $r = 0$ and $\sigma_{\mathrm{modR}}(z; \beta) = (r + \beta)e^{i\vartheta}$ on the active region $\{r > 0, \; r + \beta > 0\}$.

**Remark (value at $z = 0$ and relation to common definitions).** Many papers present modReLU in the two-branch form $\sigma(z; \beta) = (|z| + \beta) z/|z|$ if $|z| + \beta > 0$ and $0$ otherwise, without separately specifying the value at $z = 0$ where $z/|z|$ is undefined (2; 20). Our definition (1) agrees with this two-branch form for all $z \neq 0$; the only difference is that we explicitly set $\sigma_{\mathrm{modR}}(0; \beta) = 0$ to make the activation a total function (matching the usual "avoid division by zero" convention in implementations). For $\beta > 0$, modReLU cannot be made continuous at the origin under any total definition, since the directional limit depends on $z/|z|$. This convention is used throughout, and it reappears in our semi-algebraic encoding of the gate as an explicit "zero" branch; this contributes only $O(1)$ additional atomic predicates per neuron and therefore does not affect the asymptotic description complexity or the resulting $O(W^2 \log W)$ upper bound. Finally, the lower bound argument does not hinge on the value at $z = 0$: we fix $\beta_0 < 0$, for which the restriction to the real line is continuous at $0$ and the piecewise-linear structure used to invoke (11; 4) is unaffected by any choice of $\sigma_{\mathrm{modR}}(0; \beta_0)$.

For binary classification we use the sign of the real part:

$$h_\theta(x) = \mathrm{sign}\big(\Re(F_\theta(x))\big) \in \{+1, -1\}. \tag{2}$$

Via the identification $\Psi : \mathbb{R}^{2d} \to \mathbb{C}^d$, we view $h_\theta$ as a classifier on $\mathbb{R}^{2d}$ by setting, for $\tilde{x} \in \mathbb{R}^{2d}$,

$$h_\theta(\tilde{x}) = \mathrm{sign}\big(\Re(F_\theta(\Psi(\tilde{x})))\big). \tag{3}$$

Let $\mathcal{H}_{W,L}^{\mathrm{modR}}$ denote the set of all classifiers of the form (3) obtained by varying $\theta \in \mathbb{R}^W$ for a fixed architecture $(n_0, \dots, n_L)$.

## 3 VC-dimension upper bound

**Proof strategy.** We encode the forward pass of the network using auxiliary real variables for each hidden unit (real/imaginary parts of pre-activations and post-activations, and a radius variable enforcing $r^2 = u_R^2 + u_I^2$). With these variables, each affine layer relation and the modReLU gate can be written as a Boolean combination of polynomial equalities and inequalities of constant degree. The resulting decision set $\{(\theta, \tilde{x}) : \Re(F_\theta(\Psi(\tilde{x}))) \geq 0\}$ is then the projection of a semi-algebraic set defined by a single existential quantifier block over the auxiliary variables. By Tarski–Seidenberg and quantitative one-block quantifier elimination bounds, this projection admits a quantifier-free semi-algebraic description with controlled number and degrees of polynomials. We then apply the Goldberg–Jerrum theorem to convert this description complexity into the stated VC-dimension upper bound.

**Theorem 2 (VC-dimension of modReLU networks)** *There exists a universal constant $C > 0$ such that for every choice of layer widths $(n_0, \dots, n_L)$, letting $W \geq 2$ denote the total number of trainable real parameters (including real/imaginary parts of all weights and biases and all modReLU bias parameters), the VC-dimension of $\mathcal{H}_{W,L}^{\mathrm{modR}}$ satisfies*

$$\mathrm{VCdim}\big(\mathcal{H}_{W,L}^{\mathrm{modR}}\big) \leq C W^2 \log W.$$

*In particular, for any fixed depth $L$, $\mathrm{VCdim}(\mathcal{H}_{W,L}^{\mathrm{modR}}) = O(W^2 \log W)$ uniformly over architectures with $W$ parameters.*

### 3.1 Semi-algebraic representation of the decision map

Fix an architecture $(n_0, \dots, n_L)$ and write $\tilde{x} \in \mathbb{R}^{2d}$ for the real input and $\theta \in \mathbb{R}^W$ for the trainable parameters. Recall our sign convention $\mathrm{sign}(t) = +1$ for $t \geq 0$ and $-1$ for $t < 0$, so

$$h_\theta(\tilde{x}) = +1 \iff \Re(F_\theta(\Psi(\tilde{x}))) \geq 0.$$

**Auxiliary variables and constraints.** We introduce auxiliary real variables encoding a forward pass. For each hidden layer $\ell = 1, \ldots, L-1$ and unit $j = 1, \ldots, n_\ell$, we introduce

$$u_{j,R}^{(\ell)}, u_{j,I}^{(\ell)} \quad \text{(pre-activation)}, \qquad z_{j,R}^{(\ell)}, z_{j,I}^{(\ell)} \quad \text{(post-activation)}, \qquad r_j^{(\ell)} \quad \text{(radius)}.$$

(We treat $z^{(0)} = \Psi(\tilde{x})$ as fixed by $\tilde{x}$.) Let $Y$ denote the full collection of these auxiliary variables. Then

$$|Y| \;=\; 5 \sum_{\ell=1}^{L-1} n_\ell.$$

Moreover, since $n_{\ell-1} \geq 1$ for all $\ell$ and the parameter count includes the real and imaginary parts of all complex weights, we have

$$W \;\geq\; 2 \sum_{\ell=1}^{L-1} n_\ell n_{\ell-1} \;\geq\; 2 \sum_{\ell=1}^{L-1} n_\ell,$$

and hence $|Y| \leq \frac{5}{2} W$, i.e., $|Y| = O(W)$ with an absolute constant.

We impose the following constraints in the variables $(\theta, \tilde{x}, Y)$. Each atomic predicate is a polynomial equality or inequality of degree at most 2. The only non-affine ingredient is the modReLU nonlinearity: it is captured by introducing the radius variables $r_j^{(\ell)}$ and enforcing a three-branch gate (zero/inactive/active), including the $u_j^{(\ell)} = 0$ case required by the standard convention $\sigma_{\mathrm{modR}}(0; \beta) = 0$ (20).

- *Complex affine maps.* For each $\ell = 1, \ldots, L-1$ and $j = 1, \ldots, n_\ell$, we enforce the real and imaginary parts of

$$u_j^{(\ell)} \;=\; \sum_{k=1}^{n_{\ell-1}} W_{jk}^{(\ell)} z_k^{(\ell-1)} + b_j^{(\ell)}$$

by two polynomial equalities. Written in real coordinates, these constraints are bilinear in the parameter variables (real/imaginary parts of $W_{jk}^{(\ell)}$ and $b_j^{(\ell)}$) and the activation variables (real/imaginary parts of $z_k^{(\ell-1)}$), hence have degree 2.

- *modReLU gate (including the $u = 0$ case).* Fix a hidden unit $(\ell, j)$ with bias parameter $\beta_j^{(\ell)} \in \mathbb{R}$ (a component of $\theta$). We first enforce the radius relation

$$r_j^{(\ell)} \geq 0, \qquad (r_j^{(\ell)})^2 = (u_{j,R}^{(\ell)})^2 + (u_{j,I}^{(\ell)})^2.$$

We then encode $z_j^{(\ell)} = \sigma_{\mathrm{modR}}(u_j^{(\ell)}; \beta_j^{(\ell)})$ (with the convention $\sigma_{\mathrm{modR}}(0; \beta) = 0$, cf. (1) and (20)) by the disjunction of the following polynomial conditions:

$$\text{(zero)} \; r_j^{(\ell)} = 0, \quad z_{j,R}^{(\ell)} = 0, \quad z_{j,I}^{(\ell)} = 0;$$

$$\text{(inactive)} \; r_j^{(\ell)} > 0, \quad r_j^{(\ell)} + \beta_j^{(\ell)} \leq 0, \quad z_{j,R}^{(\ell)} = 0, \quad z_{j,I}^{(\ell)} = 0;$$

$$\text{(active)} \; r_j^{(\ell)} > 0, \quad r_j^{(\ell)} + \beta_j^{(\ell)} \geq 0,$$
$$z_{j,R}^{(\ell)} r_j^{(\ell)} = (r_j^{(\ell)} + \beta_j^{(\ell)}) u_{j,R}^{(\ell)}, \quad z_{j,I}^{(\ell)} r_j^{(\ell)} = (r_j^{(\ell)} + \beta_j^{(\ell)}) u_{j,I}^{(\ell)}.$$

The "zero" branch is essential: when $u_j^{(\ell)} = 0$ (hence $r_j^{(\ell)} = 0$), modReLU outputs 0 even if $\beta_j^{(\ell)} > 0$ by the standard convention in (20). Introducing this branch ensures the existential encoding matches the actual network computation for all $(u_j^{(\ell)}, \beta_j^{(\ell)})$.

- *Output and decision.* Write the output as $F_\theta(x) = W^{(L)} z^{(L-1)} + b^{(L)}$ and enforce the decision event by the single polynomial inequality

$$\Re(F_\theta(\Psi(\tilde{x}))) \geq 0,$$

again expressed in real coordinates (degree $\leq 2$ in $(\theta, \tilde{x}, Y)$).

The total number of atomic predicates across all layers is $s_0 = O(W)$, and the maximum degree is $\delta = 2$.

**From constraints to a quantifier-free description.** Define

$$G := \{(\theta, \tilde{x}) \in \mathbb{R}^{W+2d} : h_\theta(\tilde{x}) = +1\}.$$

Let $\widetilde{G} \subset \mathbb{R}^{W+2d+|Y|}$ be the set of all $(\theta, \tilde{x}, Y)$ satisfying the constraints above. Then $\widetilde{G}$ is semi-algebraic by definition, and

$$(\theta, \tilde{x}) \in G \iff \exists Y \in \mathbb{R}^{|Y|} \text{ such that } (\theta, \tilde{x}, Y) \in \widetilde{G}.$$

Hence $G$ is the projection of the semi-algebraic set $\widetilde{G}$. By the Tarski–Seidenberg theorem (quantifier elimination for real closed fields), projections of semi-algebraic sets are semi-algebraic, and therefore $G$ is itself semi-algebraic (19; 5).

**Quantitative bounds for eliminating one existential block.** The equivalence above exhibits $G$ as the set defined by a first-order formula with a *single existential quantifier block* over the auxiliary variables $Y$. While Tarski–Seidenberg guarantees existence of some quantifier-free semi-algebraic description of $G$, we also need explicit control of the *description complexity* (how many polynomials appear and what degrees they have), because this is what will enter the Goldberg–Jerrum VC-dimension bound in Theorem 1. Quantitative quantifier-elimination results provide such control for one-block formulas: starting from a quantifier-free description of $\widetilde{G}$ using $s_0$ input polynomials of degree at most $\delta$, they bound the number $s$ and maximum degree $d$ of the polynomials appearing in an equivalent quantifier-free description after eliminating the $k_1$ quantified variables, with a singly-exponential dependence on the total number of variables.

More concretely, quantitative quantifier-elimination bounds apply here because the defining formula for $\widetilde{G}$ has a single existential block, uses at most $s_0 = O(W)$ input polynomials of degree $\leq \delta = 2$, and ranges over

$$k_0 := W + 2d \quad \text{free variables} \qquad \text{and} \qquad k_1 := |Y| = O(W) \quad \text{quantified variables}.$$

In particular, by (5, Theorem 14.16) (see also the discussion of singly-exponential bounds for one-block elimination therein), there exist absolute constants $C_1, C_2 > 0$ and an equivalent quantifier-free formula defining $G$ that involves at most

$$s \leq (s_0 \delta)^{C_1(k_0+k_1)} \qquad \text{polynomials of degree at most} \qquad d \leq \delta^{C_2(k_0+k_1)}.$$

Since $n_L = 1$, any fully connected architecture satisfies $W \geq 2d + 2$ (already for $L = 1$ one has $W = 2d+2$), so $k_0 + k_1 = O(W)$ and $\log(sd) = O(W \log W)$.

**Lemma 1 (Semi-algebraic decision map)** *There exist integers $s \geq 1$, $d \geq 1$ and polynomials $p_1, \ldots, p_s \in \mathbb{R}[U_1, \ldots, U_W, X_1, \ldots, X_{2d}]$ of degree at most $d$ such that for all $\theta \in \mathbb{R}^W$ and $\tilde{x} \in \mathbb{R}^{2d}$,*

$$h_\theta(\tilde{x}) = +1 \iff \Phi\big(\text{sign}(p_1(\theta, \tilde{x})), \ldots, \text{sign}(p_s(\theta, \tilde{x}))\big) = \text{true},$$

*for a fixed Boolean formula $\Phi$ independent of $\theta, \tilde{x}$. Furthermore, one may choose $s, d$ so that for absolute constants $c_0, c_1 > 0$,*

$$s \leq (c_0 W)^{c_1 W} \qquad \text{and} \qquad d \leq 2^{c_1 W}.$$

*In particular, $\log(sd) = O(W \log W)$ uniformly over architectures with $W$ parameters.*

**Proof.** By construction, $\widetilde{G} \subset \mathbb{R}^{W+2d+|Y|}$ is semi-algebraic and is defined by a quantifier-free formula that uses $s_0 = O(W)$ polynomial (in)equalities of degree at most $\delta = 2$ (including the explicit "zero" branch for the modReLU gate, ensuring correctness when $u_j^{(\ell)} = 0$ and $\beta_j^{(\ell)} > 0$). The set $G$ is the projection of $\widetilde{G}$ onto the $(\theta, \tilde{x})$-coordinates, hence is semi-algebraic by Tarski–Seidenberg (19; 5).

Applying the quantitative one-block quantifier elimination bound (5, Theorem 14.16) with $k_0 = W + 2d$ free variables, $k_1 = |Y|$ quantified variables, $s_0$ input polynomials, and degree $\delta = 2$, we obtain an equivalent quantifier-free formula for $G$ involving at most $s \leq (s_0 \delta)^{C_1(k_0+k_1)}$ polynomials of degree at most $d \leq \delta^{C_2(k_0+k_1)}$.

Finally, any quantifier-free Boolean combination of atomic comparisons $q(\theta, \tilde{x}) \bowtie 0$ with $\bowtie \in \{<, \leq, =, \geq, >\}$ can be rewritten as a Boolean formula in the values $\text{sign}(q(\theta, \tilde{x}))$ and $\text{sign}(-q(\theta, \tilde{x}))$ under our two-valued

convention (e.g., $q > 0$ is equivalent to $q \geq 0$ and not $(q = 0)$, and $q = 0$ is equivalent to $q \geq 0$ and $-q \geq 0$). This transformation increases the number of polynomials by at most a factor of 2 and does not increase degrees, yielding the representation in terms of $\text{sign}(p_i)$ as stated.

Since $W \geq 2d + 2$ and $|Y| = O(W)$, we have $k_0 + k_1 = O(W)$ and $s_0 = O(W)$, hence $s \leq (c_0 W)^{c_1 W}$ and $d \leq 2^{c_1 W}$ for absolute constants $c_0, c_1 > 0$, and therefore $\log(sd) = O(W \log W)$. □

## 3.2 Proof of the VC-dimension bound

For each $\theta \in \mathbb{R}^W$, let
$$S_\theta := \{\tilde{x} \in \mathbb{R}^{2d} : h_\theta(\tilde{x}) = +1\}.$$

By Lemma 1, there exist integers $s \geq 1$, $d \geq 1$ and polynomials $p_1, \ldots, p_s \in \mathbb{R}[U_1, \ldots, U_W, X_1, \ldots, X_{2d}]$ of degree at most $d$, together with a fixed Boolean formula $\Phi$, such that for all $\theta \in \mathbb{R}^W$ and $\tilde{x} \in \mathbb{R}^{2d}$,

$$\tilde{x} \in S_\theta \quad \Longleftrightarrow \quad \Phi\big(\text{sign}(p_1(\theta, \tilde{x})), \ldots, \text{sign}(p_s(\theta, \tilde{x}))\big) = \text{true}.$$

Thus the family $\{S_\theta\}_{\theta \in \mathbb{R}^W}$ is a semi-algebraic concept class of the form required by the Goldberg–Jerrum theorem (10) (see also (1, Theorem 8.3) and (5, Theorem 7.38)). Applying Theorem 1 with parameter dimension $W$ yields
$$\text{VCdim}(\mathcal{H}_{W,L}^{\text{modR}}) = \text{VCdim}(\{S_\theta\}_{\theta \in \mathbb{R}^W}) \leq C W \log(sd),$$

for a universal constant $C > 0$.

It remains to bound $\log(sd)$ in terms of $W$. Lemma 1 provides such a bound uniformly over architectures: the quantifier-elimination estimate used there depends on the number of free variables $k_0 = W + 2d$ and quantified variables $k_1 = |Y|$, but these are both $O(W)$ for the present class. Indeed, since $n_L = 1$, even in the shallow case $L = 1$ we have
$$W = 2n_1 n_0 + 2n_1 = 2d + 2,$$

and for $L \geq 2$ the parameter count only increases. Hence for all architectures considered here,

$$W \geq 2d + 2, \tag{4}$$

so in particular $2d \leq W$ and $k_0 = W + 2d \leq 2W$. Together with $|Y| = O(W)$ (as shown in the construction of Lemma 1), we obtain $k_0 + k_1 = O(W)$, and Lemma 1 yields $\log(sd) = O(W \log W)$ with absolute constants (via the quantitative one-block elimination bounds of (5, Theorem 14.16)).

Therefore,
$$\text{VCdim}(\mathcal{H}_{W,L}^{\text{modR}}) \leq C W \log(sd) \leq C' W^2 \log W$$

for a universal constant $C' > 0$, completing the proof of Theorem 2. □

**Comparison with classical real-network VC-dimension proofs.** For deep real networks with piecewise-linear activations (e.g., ReLU), the nearly-tight $\Theta(WL \log W)$ bounds are obtained by counting activation sign patterns/affine regions layer-by-layer and controlling how many distinct dichotomies the network can implement (11; 4). For modReLU networks, when written in real coordinates $z = x + iy$ introduces normalization by $|z| = \sqrt{x^2 + y^2}$, so the resulting map is not piecewise-polynomial in $(x, y)$ in the sense exploited by those region-counting arguments. Our proof instead proceeds by lifting the computation to a polynomial constraint system (Section 3, Lemma 1) and then projecting out auxiliary variables using quantifier elimination (19; 5). The quantitative blow-up incurred by this elimination step is reflected in the bound $\log(sd) = O(W \log W)$ and is precisely what introduces the additional factor of $W$ in the upper bound $O(W^2 \log W)$.

**Why the standard $O(WL \log W)$ proofs for real networks do not apply as a black box after $\mathbb{C}^d \simeq \mathbb{R}^{2d}$.** The nearly-tight upper bounds of (11; 4) are proved for networks whose activation is *piecewise linear* (and, in earlier work, more generally *piecewise polynomial*) with a *constant* number of pieces. A key step in these proofs is that, once one fixes the activation "gating" pattern (i.e., which linear piece each

hidden unit operates on for a given input), the network output restricted to that region becomes an *exact* polynomial expression of controlled degree in the underlying variables (equivalently: the model reduces to a composition of affine maps with a fixed finite branch structure). This permits bounding the number of distinct sign patterns realized on a finite sample via classical sign-pattern bounds for polynomial families (e.g., Warren-type inequalities), together with a union bound over admissible gating patterns; see (11; 4) for the implementation of this strategy in the piecewise-linear setting and its relationship to the piecewise-polynomial case.

After identifying $\mathbb{C}^d \simeq \mathbb{R}^{2d}$, modReLU does *not* fall into this framework: on its active branch it contains the normalization term $u/|u|$, i.e., division by $\sqrt{u_R^2 + u_I^2}$ in real coordinates, so the resulting real-coordinate map is not piecewise-polynomial in the input variables (nor does it admit an exact finite "linear-region" description as required by the above gating-based arguments). Consequently, the polynomial sign-pattern machinery used to obtain $O(WL \log W)$ bounds cannot be invoked directly for modReLU by a simple realification argument.

One might ask whether this obstacle can be bypassed by approximating the realified modReLU map by piecewise polynomials. However, VC-dimension concerns *exact* dichotomies on arbitrary finite sets, and points in a shattered set may lie arbitrarily close to a decision boundary. Without an additional *margin* assumption, uniform function approximation does not guarantee preservation of the induced labels $\operatorname{sign}(f(x))$ on such adversarial point sets. Moreover, any approximation scheme would typically require the number of pieces and/or polynomial degrees to grow with the target tolerance, and then the resulting VC-dimension bounds would depend on this growing description complexity rather than yielding a clean $O(WL \log W)$ bound at fixed $W$.

For these reasons, our upper bound instead proceeds by an *exact* semi-algebraic lifting: we introduce auxiliary radius variables $r$ with the polynomial constraint $r^2 = u_R^2 + u_I^2$ so that the modReLU gate and the full forward pass admit a bounded-degree polynomial constraint system, and we then control the projection to $(\theta, \tilde{x})$ via one-block quantifier elimination before applying Goldberg–Jerrum.

### 3.3 Lower bounds and comparison

The upper bound of Theorem 2 is obtained via a semi-algebraic encoding and is not expected to be tight. In the opposite direction, we obtain a lower bound by observing that $\mathcal{H}_{W,L}^{\mathrm{modR}}$ contains, as a parameter-restricted subclass, standard real-valued depth-$L$ networks with a fixed piecewise-linear activation. The result below is therefore an *inherited corollary* of the nearly-tight VC-dimension lower bounds for real networks proved in (11; 4).

**Proposition 1 (Lower bound (inherited from real networks))** *There exists a universal constant $c > 0$ such that the following holds. For all integers $L \geq 2$ and all $W$ sufficiently large compared to $L$, there exists an architecture of depth $L$ with at most $W$ trainable real parameters (counting real and imaginary parts of weights and biases, as well as all modReLU bias parameters) for which*

$$\operatorname{VCdim}\left(\mathcal{H}_{W,L}^{\mathrm{modR}}\right) \;\geq\; c\,WL \log\!\left(\frac{W}{L}\right).$$

*In particular, for suitable depth-$L$ architectures, the VC-dimension of modReLU complex networks grows at least on the order of $WL \log(W/L)$.*

**Proof (via a real-valued subclass).** Fix the non-degenerate constant $\beta_0 := -1$ and consider the restriction of modReLU to the real line:

$$\phi_{\beta_0}(x) \;:=\; \Re\!\left(\sigma_{\mathrm{modR}}(x + i0; \beta_0)\right), \qquad x \in \mathbb{R}.$$

Since $\beta_0 < 0$, the activation is continuous at $x = 0$ on the real line, and the precise convention for $\sigma_{\mathrm{modR}}(0; \beta_0)$ does not affect the piecewise-linear structure used in the lower-bound argument. By the definition (1) and the convention $\sigma_{\mathrm{modR}}(0; \beta_0) = 0$ (as in (2; 20)), $\phi_{\beta_0}$ is piecewise-linear on $\mathbb{R}$ with a constant number of linear pieces (indeed, at most three). Hence the real-network lower bounds for fixed piecewise-linear activations

apply: by (11; 4), there exists a universal constant $c_0 > 0$ such that for all $L$ and all $P$ sufficiently large compared to $L$, there exists a depth-$L$ *real* network class $\mathcal{H}_{P,L}^{\text{real}}(\phi_{\beta_0})$ with at most $P$ trainable real parameters and activation $\phi_{\beta_0}$ satisfying

$$\text{VCdim}\big(\mathcal{H}_{P,L}^{\text{real}}(\phi_{\beta_0})\big) \;\geq\; c_0 \, P L \log\!\Big(\frac{P}{L}\Big). \tag{5}$$

We now embed such a real network as a subclass of our complex modReLU networks. Consider the complex-valued architecture with the same layer widths and impose the restrictions:

- inputs are restricted to $\mathbb{R}^d \subset \mathbb{C}^d$ (identified with $x + i0$),

- all complex weights and biases are constrained to be real (imaginary parts fixed to 0),

- all modReLU bias parameters are fixed to the constant value $\beta_0$ (i.e., $\beta_j^{(\ell)} = \beta_0$ for all hidden units).

Under these restrictions, the complex network computes exactly the same real-valued function on $\mathbb{R}^d$ as the corresponding real network with activation $\phi_{\beta_0}$, and the induced classifier $\tilde{x} \mapsto \text{sign}(\Re(F_\theta(\Psi(\tilde{x}))))$ coincides with the real-network classifier $\text{sign}(f(x))$ on inputs $x \in \mathbb{R}^d$.

Let $P$ denote the number of *free* real parameters (weights and biases) in the embedded real network. Consider the corresponding complex architecture (same widths) used to realize this subclass. In the ambient count $W$ for complex modReLU networks we include: (i) both real and imaginary parts of all complex weights and biases, and (ii) all modReLU bias parameters. Under the restriction "imaginary parts $= 0$" and "$\beta_j^{(\ell)} = \beta_0$", the number of free parameters is precisely the number of real weights and biases, namely $P$. Since each such parameter corresponds to a complex weight/bias coordinate that contributes two real degrees of freedom in the ambient parameterization, we have

$$P \;\leq\; \frac{W}{2}. \tag{6}$$

(Indeed, for this fixed architecture the ambient parameter count can be written as $W = 2P + P_\beta$, where $P_\beta = \sum_{\ell=1}^{L-1} n_\ell$ counts the modReLU bias parameters; in particular $W \geq 2P$.)

Consequently, $\mathcal{H}_{W,L}^{\text{modR}}$ contains a subclass with VC-dimension at least that of $\mathcal{H}_{P,L}^{\text{real}}(\phi_{\beta_0})$, and by (5) we obtain

$$\text{VCdim}\big(\mathcal{H}_{W,L}^{\text{modR}}\big) \;\geq\; \text{VCdim}\big(\mathcal{H}_{P,L}^{\text{real}}(\phi_{\beta_0})\big) \;\geq\; c_0 \, P L \log\!\Big(\frac{P}{L}\Big).$$

Finally, we choose an architecture whose total parameter count is at most $W$ and for which the embedded real subclass has $P$ on the order of $W$ up to constants. Concretely, take any depth-$L$ fully connected real architecture achieving (5) with $P = \lfloor W/3 \rfloor$ trainable parameters. For the corresponding complex architecture (same widths), the ambient parameter count satisfies $W_{\text{arch}} = 2P + P_\beta$, where $P_\beta = \sum_{\ell=1}^{L-1} n_\ell$ counts the modReLU bias parameters. Choosing the widths so that $P_\beta \leq P$ (which is compatible with standard constructions used to realize (5)), we obtain $W_{\text{arch}} \leq 3P \leq W$. Substituting $P = \lfloor W/3 \rfloor$ gives

$$\text{VCdim}\big(\mathcal{H}_{W,L}^{\text{modR}}\big) \;\geq\; c_0 \Big\lfloor \frac{W}{3} \Big\rfloor L \log\!\Big(\frac{\lfloor W/3 \rfloor}{L}\Big) \;\geq\; c \, W L \log\!\Big(\frac{W}{L}\Big),$$

for a (possibly smaller) universal constant $c > 0$, provided $W$ is sufficiently large compared to $L$. This proves the proposition. $\qquad\square$

Combining Proposition 1 with Theorem 2, we obtain for suitable depth-$L$ architectures and large $W$:

$$c \, W L \log\!\Big(\frac{W}{L}\Big) \;\lesssim\; \text{VCdim}\big(\mathcal{H}_{W,L}^{\text{modR}}\big) \;\lesssim\; W^2 \log W.$$

Thus the VC-dimension grows at least linearly in both $W$ and $L$, and at most quadratically in $W$ up to a logarithmic factor. Narrowing this gap—for instance by sharpening the upper bound towards the $W L \log W$ behavior known for real piecewise-linear networks (11; 4)—is left as an open problem.

# 4 Discussion and related work

We have shown that the VC-dimension of fully connected complex-valued networks with modReLU activation and $W$ trainable real parameters is at most on the order of $W^2 \log W$, uniformly over architectures of a given size. The proof has two main ingredients: (i) after introducing auxiliary variables for complex magnitudes, the network computation and the decision event $\Re(F_\theta(\Psi(\tilde{x}))) \geq 0$ admit an existential semi-algebraic description, and (ii) semi-algebraic concept classes with bounded description complexity have VC-dimension bounded by $O(W \log(sd))$ via the Goldberg–Jerrum theorem (10) (see also (1)). Quantitative bounds for one-block quantifier elimination (5) then control $\log(sd)$ in terms of $W$, while the Tarski–Seidenberg theorem guarantees that the projection remains semi-algebraic (19), yielding the stated $O(W^2 \log W)$ upper bound.

**Special cases (shallow and linear models).** It is instructive to consider the extreme shallow case $L = 1$, where the network reduces to a single complex affine functional with sign output,

$$h_\theta(\tilde{x}) = \text{sign}(\Re(\langle w, \Psi(\tilde{x})\rangle + b)), \qquad w \in \mathbb{C}^d, \ b \in \mathbb{C}.$$

Under the identification $\mathbb{C}^d \cong \mathbb{R}^{2d}$, the map $\tilde{x} \mapsto \Re(\langle w, \Psi(\tilde{x})\rangle + b)$ is a real affine functional on $\mathbb{R}^{2d}$, and therefore $\mathcal{H}_{W,1}^{\text{modR}}$ is a class of halfspaces in $\mathbb{R}^{2d}$. In particular, its VC-dimension is $2d + 1$, which is linear in the ambient parameter count $W = 2d + 2$ (1; 21). More generally, if all activations are linear (so the network is globally affine in the input), then regardless of depth the induced classifier is again a halfspace in $\mathbb{R}^{2d}$, and the VC-dimension is linear in the effective input dimension (and hence linear in $W$ for fully connected architectures).

**Relation to nearly-tight bounds for real networks.** For real-valued networks, sharp VC-dimension and pseudo-dimension bounds are known for broad classes of piecewise-linear architectures: in particular, the nearly-tight $\Theta(WL \log W)$ behavior for depth-$L$ networks with $W$ parameters is established in (11; 4). Our lower bound is inherited by embedding such real networks as a parameter-restricted subclass of modReLU complex networks (Section 3, Proposition 1). In contrast, our upper bound is *not* a black-box consequence of (11; 4). The upper-bound techniques in (11; 4) rely on the fact that, with piecewise-linear (more generally, piecewise-polynomial) activations having a constant number of pieces, the network computes a piecewise-polynomial function of the input on a partition induced by activation gates; this enables a layer-by-layer counting of regions/sign patterns without invoking general projection or quantifier-elimination machinery. After identifying $\mathbb{C}^d \cong \mathbb{R}^{2d}$, modReLU does not fit into this framework: writing $z = x + iy$ and $r = \sqrt{x^2 + y^2}$, the active branch contains the normalization $z/|z| = (x + iy)/r$ and hence factors of $1/\sqrt{x^2 + y^2}$, so the resulting real-coordinate map is not piecewise-polynomial in $(x, y)$ in the sense required by these arguments. For this reason, the existing $\Theta(WL \log W)$ proofs for piecewise-linear networks cannot be applied directly to obtain comparable upper bounds for modReLU complex networks. Our approach instead proceeds by lifting the computation to an existential system of polynomial constraints (introducing auxiliary variables $r$ with $r^2 = x^2 + y^2$), and then projecting out these auxiliary variables using Tarski–Seidenberg/quantifier elimination (19; 5). Combined with the Goldberg–Jerrum bound for semi-algebraic concept classes (10), this yields our $O(W^2 \log W)$ VC-dimension upper bound.

**Can complex networks exceed real-valued capacity?** A complex-valued network induces a real-valued classifier on $\mathbb{R}^{2d}$ via the identification $\mathbb{C}^d \simeq \mathbb{R}^{2d}$, and each complex affine map $u = Wz + b$ with $W = A + iB$ can be written as a real affine map with the structured block matrix $\begin{bmatrix} A & -B \\ B & A \end{bmatrix}$ acting on $(\Re z, \Im z)$. Thus, for a fixed *real* parameter budget $W$, the modReLU hypothesis class considered here is (after identification) a *subclass* of real networks with $W$ free real parameters but additional algebraic constraints (weight-tying induced by complex arithmetic). In this sense, when capacity is measured as a function of $W$, complex networks cannot have larger VC-dimension than an unconstrained real network family of comparable size; any advantage of complex representations is better viewed as *parameter-efficiency* or inductive bias rather than strictly larger worst-case combinatorial capacity. Although the decision uses $\Re(F_\theta)$, the imaginary part still affects hidden-layer activations and can change $\Re(F_\theta)$, so this does not reduce the model to a purely real network. Finally, the same semi-algebraic lifting applies to other real decision statistics of a complex output (e.g., $\text{sign}(\Im(F_\theta))$, thresholding $|F_\theta|$, or phase regions), and a systematic comparison of such variants to classical real architectures remains an open direction.

**Why sharpening the upper bound is technically difficult.** The gap between our $O(W^2 \log W)$ upper bound and the $\Theta(WL \log W)$ behavior known for real piecewise-linear networks arises from the use of generic projection/quantifier-elimination bounds. For ReLU-type activations, one can analyze the network layer-by-layer and count the number of activation sign patterns (or affine regions) induced by hyperplane arrangements, leading to sharp control of the shatter function and VC-dimension (11; 4). For modReLU, the normalization $z/|z|$ introduces algebraic structure (involving $\sqrt{x^2 + y^2}$) that obstructs a direct reduction to counting finitely many affine regions in the $(x, y)$-plane. Our proof repairs this by lifting to an existential semi-algebraic description using auxiliary radius variables and then projecting them out using quantifier elimination (Tarski–Seidenberg) (19; 5). Quantitative elimination is necessarily worst-case and can introduce a blow-up in the number and degrees of polynomials describing the projected set; in our argument this manifests as the bound $\log(sd) = O(W \log W)$ (Lemma 1), which, when combined with Goldberg–Jerrum (10), yields the additional factor of $W$ in $O(W^2 \log W)$. Improving the upper bound towards $O(WL \log W)$ would therefore require bypassing generic quantifier elimination and exploiting finer compositional structure of modReLU and complex multiplication to control the number of distinct dichotomies more directly.

**Consequences of bounded VC-dimension.** A bound on $\mathrm{VCdim}(\mathcal{H}_{W,L}^{\mathrm{modR}})$ controls classical combinatorial quantities such as the growth function (shatter coefficient), via Sauer–Shelah type inequalities (21; 1). It also yields standard distribution-free sample-complexity guarantees for uniform convergence and ERM-style learning: informally, achieving small generalization error requires a number of samples scaling on the order of $\mathrm{VCdim}(\mathcal{H}_{W,L}^{\mathrm{modR}})$ up to logarithmic factors (21; 1; 22). While such worst-case guarantees are typically loose for modern overparameterized training, they provide a principled capacity baseline for comparing complex-valued architectures to their real-valued counterparts.

**Outlook.** A natural open problem is to sharpen the $O(W^2 \log W)$ upper bound towards the $O(WL \log W)$ behavior known for real piecewise-linear networks (11; 4), potentially by avoiding generic quantifier-elimination bounds and instead exploiting finer structure of modReLU and complex multiplication. Another direction is to extend the semi-algebraic approach to other complex activations beyond modReLU, and to other architectures such as convolutional, recurrent, or attention-based complex-valued models, where the interaction of weight sharing, complex arithmetic, and activation structure may lead to different capacity scalings. Finally, it would be interesting to compare VC-type bounds to norm-based generalization analyses for CVNNs, such as spectral-complexity bounds (7), and to study pseudo-dimension for complex-valued regression settings (1; 22).

**Limitations.** Our analysis is restricted to fully connected feedforward architectures with modReLU activation, and we do not track constants or lower-order terms in the VC-dimension bounds. The semi-algebraic encoding relies on quantifier elimination (5; 19), which is computationally heavy; here it is used purely as a theoretical tool. The resulting $O(W^2 \log W)$ upper bound is likely not optimal, as suggested by the nearly-tight $O(WL \log W)$ bounds for real piecewise-linear networks (11; 4). We also do not address data-dependent or norm-based complexity measures for complex-valued networks, nor do we consider convolutional, recurrent, or attention-based architectures.

**Novelty and practical relevance.** We emphasize that the principal novelty of this work is methodological rather than a direct application of known results: while the Goldberg–Jerrum theorem and quantifier-elimination machinery are standard tools, the challenge lies in showing that modReLU networks—whose real-coordinate representation is not piecewise-polynomial—admit a semi-algebraic encoding with description complexity controlled purely in terms of the parameter count $W$. This encoding step is specific to the algebraic structure of modReLU (and, more broadly, to activations involving $|z|$) and does not follow from any black-box reduction to existing real-network VC-dimension theory. From a practical standpoint, the resulting bounds provide the first principled capacity baseline for modReLU networks: they give distribution-free sample-complexity guarantees via standard VC-theory (21; 1) and enable a direct comparison of the combinatorial complexity of complex-valued architectures against their real-valued counterparts, complementing the norm-based generalization analyses of (7). While worst-case VC-dimension bounds are known to be loose for modern overparameterized regimes, they remain the standard combinatorial notion of capacity in learning theory and serve as a necessary theoretical foundation for the complex-valued setting.

**Reproducibility Statement**

This is a theoretical paper. All claims are stated as formal theorems, lemmas, or propositions, with proofs provided in the main text or as proof sketches that rely on standard results from real algebraic geometry (5; 19) and statistical learning theory (21; 1). No datasets, code, or hyperparameters were used.

**Broader Impact Statement.** This work is purely theoretical and studies the combinatorial capacity of a class of complex-valued neural networks. We do not anticipate direct negative societal impacts beyond those already associated with general advances in machine learning theory. Any downstream impact will arise only through applications that use complex-valued networks; in those settings, standard considerations around fairness, robustness, and potential misuse of models still apply.

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
