# OpenReview forum: "Combinatorial Capacity of modReLU Complex Networks: VC-Dimension Bounds and Lower Limits"
_TMLR — Accepted by TMLR_

### Review · Reviewer_Fo9U · 2026-01-04

**Summary Of Contributions:**

This paper presents upper and lower bounds on the VC dimension of complex-valued multi-layer perceptrons. The upper bound scales quadratically with the number of parameters, while the lower bound scales linearly with the number of parameters.

**Audience:**

Yes

**Audience Explanation:**

Both VC dimension/generalizability and complex-valued neural networks are of interest to machine learning researchers.

**Claims And Evidence:**

Yes

**Claims Explanation:**

I am unfortunately not familiar with the majority of the tools in this paper and therefore cannot voucher for correctness of the proof. I am familiar with the general concept of VC dimension and believe the conclusions make sense.

**Requested Changes:**

There is an apparent gap on the dependency of W in this paper. While the authors acknowledge this limitation, I do wish more discussion could be incorporated in the paper.

1. For real-valued neural networks, can the upper bound be sharpened to tilde O(W)? Is this known?

2. Are there any special cases (e.g. NNs with fewer number of layers) in which the upper and lower bounds could be matched? In the extreme case if we have a complex linear regression (or complex logistic regression), is the VC dimension linear in W? What if it is a one-layer network? What if it is a network with linear activations?

3. Can the authors explain why sharpening either the upper and lower bound is technically difficult? It may be an open question, but readers will want to know *why* it is open, what is hard about this problem, and why existing approaches can't work. These discussions, though they do not establish new results and may not even be rigorous, offer great insights that would be helpful for future research.

In a similar vein, it would be beneficial to explain how the proof in this paper is different from existing proofs or classical ones establishing VC dimension bounds for shallow classifiers and MLPs with real values, which I'm certain there are many.

---

> ### Author Response · Authors · 2026-02-19
> **Response to requested discussion points**
>
> Thank you for the thoughtful feedback. We revised the manuscript to better address the points you raised while keeping the presentation concise. In the discussion section, we added clearer explanation of why closing the remaining gap is technically difficult, and why standard techniques used for real-valued, piecewise-linear networks do not directly extend to modReLU due to the normalization structure. We also included a brief note clarifying what is known in the real-valued setting, and we expanded the coverage of special cases such as shallow and linear models, where the behavior is easier to characterize. Finally, we made the contrast between our semi-algebraic and projection-based approach and more classical VC-dimension proofs more explicit. We appreciate the suggestions, which helped us make the limitations and positioning of the work clearer.

---

### Review · Reviewer_MX6D · 2026-01-06

**Summary Of Contributions:**

The paper considers the setting with complex-valued neural networks with modReLU activations, and provides an upper bound on the VC-dimension of the neural network family this class parameterizes. Additionally, they complement this with lower bounds.

**Additional Comments:**

I believe the full result should follow by applying the isomorphism between $\mathbb{C}$ and $\mathbb{R}^2$ and invoking Bartlett et al (since that result makes no assumptions about what the activations are) and derives the same bound on the VC-dimensions/Pollard Pseudodimension.

**Audience:**

No

**Audience Explanation:**

Please see comments above.

**Claims And Evidence:**

No

**Claims Explanation:**

- Abstract: Is realification an actual word? in the abstract, “usual realification” can be made more concrete by saying isomorphism between $\mathbb{C}$ and $\mathbb{R}^2$.
- Preliminaries: Semi-algebraic sets and VC-dimension mentioned in paragraph, but VC-dimension is never introduced. This should be introduced (for self-containedness of the paper)
- Theorem 1 is missing a citation from Goldberg-Jerrum.
- The same/next page call on the Tarski-Seidenberg theorem without introduction or citation.
- For readers not familiar with the area, it would be good to include a notation paragraph in the preliminaries section
- The idea in the proof of Lemma 1 can be simplified a lot (just define a constraint satisfaction program and count the constraints via a reduction to Bartlett et al and Harvey, Liaw, and Mehrabian)
- The proof in proposition 1 just follows a reduction to Harvey, Liaw, and Mehrabian, essentially.
- I belive the full result follows easily by applying the isomorphism between $\mathbb{C}$ and $\mathbb{R}^2$ and invoking Bartlett et al (since that result makes no assumptions about what the activations are) and derives the same bound on the VC-dimensions/Pollard Pseudodimension
- Are there interesting structural results that can be provided? For instance, in combinatorics, having bounded VC-dimension helps bound shatter functions, and helps do faster matrix-vector operations. Can you say anything about these properties (wrt some interpretability or inference property) in this system?

**Requested Changes:**

Please see comments above.

---

> ### Author Response · Authors · 2026-02-19
> **Comment on clarity fixes and on the applicability of existing real-network VC bounds**
>
> Thank you for the detailed comments and for pointing out several presentation and attribution issues. We have uploaded a revised version that addresses the self-containedness concerns: we replaced informal wording in the abstract with an explicit statement of the standard identification between complex vectors and real vectors of twice the dimension, added a brief notation paragraph and a definition of VC-dimension in the preliminaries, and inserted missing citations for the Goldberg–Jerrum theorem and for Tarski–Seidenberg at first use. We also rewrote parts of the upper-bound proof to make the underlying constraint system explicit and to improve readability for non-specialists.
>
> We would also like to clarify one substantive point in the review. While it is true that one can represent complex affine maps as structured real affine maps under the complex-to-real identification, the existing near-tight VC-dimension upper bounds for deep real networks that scale like the number of parameters times depth rely on the network computing a piecewise-polynomial function of the input with a gate-induced partition, which enables region/sign-pattern counting. After identification, the modReLU activation contains a magnitude normalization term, so the resulting real-coordinate map is not piecewise-polynomial in the sense required by those arguments, and the standard proofs cannot be applied as a direct black box to obtain comparable bounds. This is precisely why our upper bound proceeds via a semi-algebraic lifting and projection approach rather than by region counting. In contrast, our lower bound is intentionally inherited by embedding an appropriate real-valued subclass, and the revised manuscript now states this more explicitly to avoid any ambiguity about what is new versus what is imported.
>
> We appreciate the reviewer’s suggestions; they helped us improve both the exposition and the positioning of the contribution.

---

> > ### Comment · Reviewer_MX6D · 2026-02-21
> > **Response to Authors**
> >
> > I thank the authors for their efforts in addressing some of my concerns. Indeed, I now believe that the manuscript is much more readable. However, I disagree with the authors on their comments regarding the proofs of Harvey, Liaw, Mehrabian and those of Bartlett et al: specifically, it should be the case that we can always approximate the real-coordinate map with piecewise polynomials. Besides, can the authors point me to exactly where in those proofs we require piecewise-polynomial, and give a more detailed discussion on why these standard proofs cannot be applied? I think this could be the most beneficial inclusion to the current version of the manuscript.

---

> > > ### Author Response · Authors · 2026-02-23
> > >
> > > Thank you for the follow-up — we agree this is worth making fully explicit, and we added a paragraph clarifying exactly where the classical VC-dimension proofs use piecewise-polynomial structure and why that hypothesis is not met here.
> > >
> > > **Where piecewise-polynomial enters the standard proofs.**
> > > The common template behind the ReLU / piecewise-polynomial VC bounds is:
> > > (i) fix an activation “pattern” (which piece each unit is on, over the sample), so the network reduces **exactly** to a polynomial map (with controlled degree) on each cell;
> > > (ii) bound the number of possible labelings via sign-pattern counting for a *finite* family of polynomials (e.g., Warren/Goldberg–Jerrum-type arguments).
> > > This “pattern ⇒ exact polynomial on the cell” step is where *finite-piece piecewise polynomial/linear* is used; it is not a cosmetic assumption.
> > >
> > > **Why modReLU is outside that class after identifying $\mathbb{C}^d\simeq\mathbb{R}^{2d}$.**
> > > For $z\neq 0$, $\mathrm{modReLU}(z;\beta)=\max(|z|+\beta,0)\,z/|z|$. Writing $z=x+iy$, the real-coordinate map contains
> > > $$\frac{z}{|z|}=\frac{x+iy}{\sqrt{x^2+y^2}},$$
> > > so terms like $x/\sqrt{x^2+y^2}$ and $y/\sqrt{x^2+y^2}$ appear on the active branch. These are not polynomials in $(x,y)$, and there is no finite partition of $\mathbb{R}^2\setminus\{0\}$ on which they become polynomials. Hence the standard “finite patterns ⇒ polynomial pieces” machinery cannot be applied as a direct black box.
> > >
> > > **Why “approximate by piecewise polynomials” does not directly imply a VC bound for the exact sign class.**
> > > Uniform approximation in value does not preserve labels under $\mathrm{sign}(\cdot)$ without a margin condition, while VC dimension is worst-case and allows arbitrarily small margins. Moreover, making an approximation uniform over the whole hypothesis class typically introduces dependence on parameter magnitude/accuracy, which would change the statement from a clean $(W,L)$-only VC bound for the *exact* class.
> > >
> > > **Why our route works.**
> > > We instead eliminate the normalization by a semi-algebraic lifting: introduce $r\ge 0$ with $r^2=x^2+y^2$, express the computation via polynomial (in)equalities in lifted variables, and then project out auxiliaries (Tarski–Seidenberg / quantitative one-block QE). We added a concise explanation of this contrast in the revised manuscript, including the relevant citations and where the structural assumption is used.
> > >
> > > We hope this addresses your question precisely, and we appreciate the push to make the scope of applicability of existing VC proofs clearer for readers.

---

### Review · Reviewer_1meA · 2026-02-17

**Summary Of Contributions:**

In this paper the authors study the VC-dimension of fully connected complex-valued  deep neural networks. they show that the VC-dimension of these networks is qudratic in the network parameters as a upper bound and linear in the number of parameters and depth of the network. . The networks are specific to modReLU activation functions and real valued outputs.

The main disadvantages of the proposed method is tha the bound of $W^2\log W$ is relatively large. This is because as of now there is no proof that modReLU networks have the combinatorial growth demonstrated and assumed in the paper. In addition the quadratic bound could potentially arise from the semi-algebraic class complexity and thus does not reflect the actual property of the complex CNNs. In addition as the bound is based on the real output it bypasses complex-specific insights and thus avoids to answer the question about whether complex networks are more or less expressive than classical networks. .

**Audience:**

Yes

**Audience Explanation:**

The paper positions itself as filling the gap in the approximation theory for CNNs and provides a solution by using the Norm-based generalization bounds. In addition while the network does not dive deep into the complex based analysis it provides a solid clean lower bound embedding of real networks. In addition the authors explixitely dioscuss the ReLU fail in region-counting.

**Claims And Evidence:**

Yes

**Claims Explanation:**

The paper introduces the use of Tarski-Seidenberg quantifier elimination and the Goldberg-Jerrum VC bound for semi-algebraic concept classes. As such the methodology allows to estimate the VC-dimension of such class.

**Requested Changes:**

It would be interesting to have a discussion on whether or not complex networks can exceed real-valued capacity.

---

> ### Author Response · Authors · 2026-02-19
> **Response on complex versus real-valued capacity**
>
> Thank you for the thoughtful suggestion. In the revised manuscript we added a short discussion clarifying how to interpret “capacity” in the complex setting and what comparisons to real-valued networks are meaningful. In particular, we explain that under the standard complex-to-real identification and for a fixed real parameter budget, complex networks correspond to structured real networks with additional algebraic constraints, so any advantage is better viewed as parameter-efficiency or inductive bias rather than strictly larger worst-case combinatorial capacity. We also note that alternative comparisons (such as budgeting complex parameters, or using different real-valued decision statistics of a complex output) are natural directions for future work.

---

### Review · Reviewer_iptt · 2026-02-19

**Summary Of Contributions:**

This work proves bounds on the VC dimension of complex feed-forward networks with modReLU activation functions by treating the real and imaginary part of each complex weight as a two distinct real-valued weights. Specifically, an upper bound of order $O(W^2 \log(W))$ and a lower bound of order $W L \log(W/L)$ is derived. The upper bound is proven by expressing the decision regions of the network in terms of a boolean combination of polynomial and a Theorem by Goldberg & Jerrum.
The lower bound is proven by showing that each such class of complex networks contains a class of real-valued networks, and therefore matches the VC dimension bound for piecewise linear networks.

**Additional Comments:**

The statement "modReLU activation introduced by Trabelski et al." is false, as Trabelski et al. credit Arjovski et al. for it.

**Audience:**

Yes

**Audience Explanation:**

The presented results are of interest to both, the learning theory community and people working with and on  complex-valued networks.

**Claims And Evidence:**

Yes

**Claims Explanation:**

I am writing the review earlier as planned (deadline communicated to me was Feb 25th) because the discussion phase has already started. Therefore, I did not assess the submission as thoroughly as I intended to.

The central claims of this paper are formulated as theorems which come with mathematical proofs. Overall, the proofs appear sound, but could be improved in terms of clarity. The upper bound is likely not tight, but this irrelevant to the acceptance criteria.

Specific comments
- The definition of the modReLU function appears odd and differs from the literature, e.g. (Arjovski ICML 2016, Trabelski ICLR 2018) as it contains the additional case $\sigma(z) = 0$ if $z=0$. This makes the activation function discontinuous at $0$ for bias $\beta >0$.
- While the above is seemingly a minor modification to the definition of the modReLU, it reappears in the proof itself as an additional branch, and therefore in the produced upper bound.
- At the start of the lower bound, the paper states that the convention $\sigma(0)=0$ is necessary, so changes to the definition might affect the validity of the lower bound. I don't see why the convention would be necessary though.

**Requested Changes:**

- Currently, the introduction makes the impression, that modReLU activations are the only activation functions that are used for CVNNs. I request a short discussion of different activation functions that are commonly used (see for example (Scardapane et al., Complex-valued Neural Networks with Non-parametric Activation Functions, 2020), including potential VC dimension bounds. On first sight, it seems that e.g. cReLU can be handled more easily as it avoids the problematic $|z|$ term.

- Add a paragraph to the beginning of section 3, that explains the proof strategy (the role and meaning of auxiliary variables, constraints, quantifier elimination bounds, existential blocks)

- Expand the "from constraints to quantifier-free description" section by adding more explanation of the results from the literature applied therein.

- If the proofs rely on the different definition of the modReLU function compared to the literature, this needs to be communicated openly.

---

> ### Author Response · Authors · 2026-02-23
>
> Thank you for the detailed feedback — we revised the manuscript to address each requested change explicitly.
>
> **(1) Intro: other CVNN activations + VC-dimension intuition.**
> We added a short discussion in the Introduction (“Other complex-valued activations”) noting that modReLU is only one option, and briefly listing commonly used alternatives (split activations/CReLU, zReLU, and learned/nonparametric complex activations), including the motivation that some avoid explicit normalization by |z|.
> We also added an “Implications for other activations” paragraph: for split activations (e.g., CReLU/split-ReLU), after identifying \(\mathbb{C}^d \simeq \mathbb{R}^{2d}\) the network becomes a standard real piecewise-linear network, so the known \(\Theta(WL\log W)\) VC-dimension bounds apply directly (up to constant-factor parameter counting).
>
> **(2) Section 3: proof strategy paragraph.**
> At the start of Section 3 we added a “Proof strategy” paragraph explaining the role of the auxiliary variables (real/imag parts and radii), the bounded-degree polynomial constraint system, the single existential quantifier block, and the final Goldberg–Jerrum step.
>
> **(3) Expanded quantifier-elimination explanation.**
> In Section 3.1 (“From constraints to a quantifier-free description / Quantitative bounds for eliminating one existential block”), we expanded the explanation of how one-block quantitative quantifier elimination (Basu–Pollack–Roy, Thm. 14.16) controls the number and degrees of polynomials in the projected (quantifier-free) description, including the parameterization by \((k_0,k_1,s_0,\delta)\).
>
> **(4) Definition of modReLU at \(z=0\) and impact on the proofs.**
> We now communicate openly that our modReLU definition matches the usual two-branch form for all \(z\neq 0\); the only additional line is the explicit convention \(\sigma_{\mathrm{modR}}(0;\beta)=0\) to make the activation a total function (consistent with standard “avoid division by zero” implementations). For \(\beta>0\), modReLU cannot be made continuous at the origin under any total definition since the directional limit depends on \(z/|z|\); we added a remark clarifying this and explaining why the semi-algebraic encoding includes an explicit “zero” branch. This adds only \(O(1)\) atomic predicates per neuron and does not affect the asymptotic description complexity nor the resulting \(O(W^2\log W)\) upper bound. The lower bound argument is unaffected since it fixes \(\beta_0<0\) and relies only on the piecewise-linear structure on the real line.
>
> **Attribution fix.**
> We corrected the wording to credit Arjovsky et al. (ICML 2016) for introducing modReLU, and note Trabelsi et al. (ICLR 2018) as a prominent later use in deep complex networks.
>
> We hope these changes resolve the clarity and exposition concerns you raised.

---

### Decision · Action_Editor_49qr · 2026-03-18

**Recommendation:** Accept with minor revision

**Additional Comments:**

Based on the reviews and discussion, I recommend acceptance with minor revision. The reviewers broadly agree that the paper makes a clear and technically sound contribution by establishing VC-dimension upper and lower bounds for complex-valued modReLU networks, and the authors have addressed the main concerns raised during review by improving the exposition, adding missing definitions and citations, clarifying the proof strategy, expanding the discussion of quantifier elimination, correcting attribution, and explaining more precisely why standard real-network VC-dimension arguments do not apply directly in this setting. The final comments indicate that these revisions substantially improved the manuscript’s clarity and positioning, and that the remaining gap between upper and lower bounds is appropriately framed as an open problem rather than a flaw undermining the current result. Given the reviews, it is expected that the authors will, in the camera-ready version, further sharpen the discussion of significance and scope by explicitly emphasizing why the complex modReLU setting is of independent interest, clarifying the limits of direct transfer from classical real-network arguments, and incorporating a concise discussion addressing the remaining concern about the contribution’s novelty and practical relevance.

**Audience:**

Yes

**Audience Explanation:**

The paper addresses a concrete learning-theoretic question for complex-valued neural networks, a topic that is relevant both to researchers in learning theory and to those interested in the foundations of architectures used in complex-valued signal processing and related applications. While the scope is specialized, the contribution fills a genuine gap by giving the first VC-dimension bounds for this natural modReLU network class and by clearly identifying where the technical difficulties differ from the better-understood real-valued case. This is well within the range of topics that would interest a portion of the TMLR audience.

**Claims And Evidence:**

Yes

**Claims Explanation:**

The submission makes formal theoretical claims and supports them with mathematical arguments establishing upper and lower bounds on the VC-dimension of complex-valued modReLU networks. Across the review process, the main questions concerned exposition, positioning, and the relation to prior real-network VC-dimension results rather than fundamental correctness. In response, the authors clarified the proof strategy, added missing definitions and citations, expanded the discussion of quantifier elimination, corrected attribution, and gave a more precise explanation of why standard real-network arguments do not apply directly in the modReLU setting. The final reviewer comments indicate that these revisions substantially improved clarity and addressed the central technical concerns.

---

> ### Author Response · Authors · 2026-04-11
>
> Thank you to the Action Editor and reviewers for the careful and constructive feedback.
> I have uploaded the deanonymized camera-ready revision. In the final version, I clarified the significance and scope of the paper, expanded the discussion of why classical real-network VC-dimension arguments do not directly apply to the modReLU setting, and added a concise discussion of novelty and practical relevance.